# LOW BIAS GRADIENT ESTIMATES FOR VERY DEEP BOOLEAN STOCHASTIC NETWORKS

## ABSTRACT

Stochastic neural networks with discrete random variables are an important class of models for their expressiveness and interpretability. Since direct differentiation and backpropagation is not possible, Monte Carlo gradient estimation techniques have been widely employed for training such models. Efficient stochastic gradient estimators, such Straight-Through and Gumbel-Softmax, work well for shallow models with one or two stochastic layers. Their performance, however, suffers with increasing model complexity.

In this work we focus on stochastic networks with multiple layers of Boolean latent variables. To analyze such networks, we employ the framework of harmonic analysis for Boolean functions. We use it to derive an analytic formulation for the source of bias in the biased Straight-Through estimator. Based on the analysis we propose *FouST*, a simple gradient estimation algorithm that relies on three simple bias reduction steps. Extensive experiments show that FouST performs favorably compared to state-of-the-art biased estimators, while being much faster than unbiased ones. To the best of our knowledge FouST is the first gradient estimator to train very deep stochastic neural networks, with up to 80 deterministic and 11 stochastic layers in our experiments.

## 1 INTRODUCTION

Stochastic neural networks with discrete latent variables have been an alluring class of models for their expressivity and interpretability, dating back to foundational work on Helmholtz machines (Dayan et al., 1995) and sigmoid belief nets (Neal, 1992). Since they are not directly differentiable, discrete random variables do not mesh well with the workhorse of modern Deep Learning, that is the backpropagation algorithm. Monte Carlo gradient estimation is an effective solution where, instead of computing the true gradients, one can sample gradients from some distribution. The sample estimates can be either biased or unbiased. Unbiased gradient estimates like score function estimators (Williams, 1992) come typically at the cost of high variance leading to slow learning. In contrast, biased gradient estimates such Straight-Through (Bengio et al., 2013), while efficient, run the risk of convergence to poor minima and unstable training. To this end several solutions have recently been proposed that either reduce variance in unbiased estimators (Mnih & Gregor, 2014; Gu et al., 2015; Tucker et al., 2017; Rezende et al., 2014; Grathwohl et al., 2017) or control bias in biased estimators (Jang et al., 2016; Maddison et al., 2016). These methods, however, have difficulty scaling up to complex neural networks with multiple stochastic layers: low-variance unbiased estimators are too expensive [1] , while the compounded bias from the continuous relaxations on multiple stochastic layers leads to poor minima. In this work we focus on biased estimators.

Our goal in this paper is a gradient estimator for Boolean random variables that works for any complex –deep or wide– neural network architecture. We resort to the term *Boolean* instead of binary to emphasize that we work directly on the Boolean space $\{-1, +1\}$, without any continuous relaxations or quantizations. With this in mind we re-purpose the framework of harmonic analysis of Boolean functions, widely used in computational learning and computational complexity theory (O'Donnell, 2014; Linial et al., 1993; Mossel et al., 2003; Mansour, 1994). We cast stochastic neural networks as Boolean functions $f(z)$ over Boolean latent variables $z$ sampled from probability

---

[1]Training a nonlinear sigmoid belief network model with two stochastic layers on MNIST with REBAR took a day and a half on GPU.

---

**Algorithm 1** FouST Gradient Estimator

---

**Require:** Parameters $\theta \in \mathbf{R}^K$, Bernoulli Representation $\{-a, a\}$, Interval Parameter $b \in [0, a]$,
Constant scaling parameter $\gamma$
  1: Sample $x_i \sim p_{\theta_i}(x_i), i = 1, ..., K$    ▷ This and all steps below have constant complexity.
  2: Sample $y_i \sim \text{Unif}(b, a)$
  3: Set $y_i := x_i * y_i$
  4: Compute ST gradient $\partial_{\theta_i} := \partial_{y_i} f(y)$
  5: Importance reweighing $\partial_{\theta_i} := 0.5/p(x_i) \cdot \partial_{\theta_i}$
  6: **return** partial gradient $\gamma \partial_{\theta_i}, i = 1, ..., K$

---

distributions $p(z)$. We then use harmonic analysis to determine that the bias in the Straight-Through gradient estimates corresponds to the weighted sum of higher-order Taylor coefficients of $f(z)$. The direct consequence is that one can *control the bias in the Straight-Through estimator by manipulating the higher-order Taylor coefficients of $f(z)$*. Building upon the harmonic analysis of existing gradient estimators, we present an algorithm, *FouST*, that admits low bias gradient estimates for Boolean latent variable models. In experiments, we were able to scale up the stochastic depth and width of neural networks, training deep stochastic residual networks with up to 80 deterministic and 11 stochastic layers with little difficulty. We summarize FouST in Algorithm 1.

With this work we make the following three contributions.

1. We introduce the framework of harmonic analysis of Boolean functions to analyze discrete stochastic neural networks and their REINFORCE and Straight-Through gradients. We show that stochastic gradients compute Fourier coefficients.

2. Based on the above harmonic analysis we present FouST – a low-bias gradient estimator for Boolean latent variables based on three bias reduction steps. As a side contribution, we show that the gradient estimator employed with DARN (Gregor et al., 2013), originally proposed for autoregressive models, is a strong baseline for gradient estimation in large and complex models with many stochastic layers.

3. We show that FouST is amenable to complex stochastic neural networks with Boolean random variables. To the best of our knowledge, *FouST is the first gradient estimate algorithm that can train very deep stochastic neural networks with Boolean latent variables*.

The practical outcome is a simple gradient estimate algorithm that can be plugged in complex stochastic neural networks with multiple layers of Boolean random variables.

## 2 HARMONIC ANALYSIS OF BOOLEAN FUNCTIONS

We consider Boolean functions on the $n$-dimensional Boolean cube, $f : \{-1, +1\}^n \to \mathbb{R}$. The setting of Harmonic Analysis for Boolean functions is the space of Boolean functions $f$ with a product probability distribution on the Boolean input, that is $p(z) = \prod_{i=1}^n p_i(z)$. We denote $p_i$ as the probability of the $i$-th dimension being one, *i.e.*, $p_i := p_i(z_i = +1)$. We denote the mean and variance of $z_i$ by $\mu_i$ and $\sigma_i$, respectively.

An example of a Boolean function in this setting is a generative neural network $f : z \mapsto y$ with a factorized latent distribution, as commonly done in representation learning (Kingma & Welling, 2013; Higgins et al., 2017). In this example, $z$ is the stochastic input – also known as the latent code in stochastic neural networks – taking only two possible values and $y$ is the output, like cross entropy loss. Often, the goal of a generative neural network is to learn or approximate the latent distribution given input data $x$, *i.e.*, $p(z|x)$, as we will also explore in the experiments.

We first introduce a few basic operations in the context of Harmonic Analysis of Boolean functions, which we shall use further on. Further necessary details are in the Appendix A. For a more comprehensive introduction, however, we refer the reader to O'Donnell (2014).

**Inner product.** The inner product of two Boolean functions $f$ and $g$ is: $\langle f, g \rangle = \mathbb{E}_{p(z)}[f(z)g(z)]$.
**Fourier expansion.** Let $S$ be any subset of dimensions of the $n$-dimensional Boolean cube, $S \subseteq [n] = \{1, ..., n\}$. Per $S$ we define a basis function, $\phi_S(z) := \prod_{i \in S} \phi_i(z_i)$, where for the empty set

$\phi_\varnothing(z) = 1$ and $\phi_i$ is the $z$-score normalized dimension, *i.e.*, $\phi_i := \frac{z_i - \mu_i}{\sigma_i}$. For example, under the uniform Bernoulli distribution for the $i$-th dimension, $p_i(z_i = +1) = p_i(z_i = -1) = 1/2$, we have $\phi_i(z_i) = z_i$.

The $2^n$ functions $\phi_S$ form an orthonormal basis for the space of Boolean functions $f$,

$$\mathbb{E}_{p(z)}[\phi_i(z_i)] = 0, \tag{1}$$
$$\mathbb{E}_{p(z)}[\phi_i(z_i)^2] = 1,$$
$$\mathbb{E}_{p(z)}[\phi_i(z_i)\phi_j(z_j)] = \mathbb{E}_{p(z)}[\phi_i(z_i)]\,\mathbb{E}_{p(z)}[\phi_j(z_j)],$$

for $i \neq j$, where the expectations compute the inner product between two Boolean functions. The last identity derives from the independence of any dimensions $i \neq j$.

We can then expand the Boolean function $f$ on the set of $2^n$ orthonormal basis functions,

$$f(z) = \sum_{S \subset [n]} \hat{f}^{(p)}(S)\phi_S(z), \tag{2}$$

also known as the $p$-biased Fourier expansion of $f$. The $\hat{f}^{(p)}(S)$ are the Fourier coefficients computed by the inverse Fourier expansion,

$$\hat{f}^{(p)}(S) = \mathbb{E}_{p(z)}[f(z)\phi_S(z)]. \tag{3}$$

That is, the inverse expansion is computed by the inner product for Boolean functions defined above. The cardinality of $S$ is the degree of the coefficient $\hat{f}^{(p)}(S)$. For instance, we have only one degree-0 coefficient, $\hat{f}^{(p)}(\varnothing)$, which equals to the expected value of $f$ under the distribution $p(z)$, $\hat{f}^{(p)}(\varnothing) = \mathbb{E}[f(z)]$ since $\phi_\varnothing = 1$. Further, we have $n$ degree-1 coefficients $\hat{f}^{(p)}(i) = \langle f, \phi_i \rangle$ and so on.

## 3 HARMONIC ANALYSIS ON BIASED STRAIGHT-THROUGH GRADIENTS

We examine Straight-Through gradient estimates using the framework of Harmonic Analysis of Boolean functions. For training the model parameters with a loss function $\mathcal{L} := \mathbb{E}_{p(z)}[f(z)]$ we want to compute the gradient $\partial_{p_i}\mathcal{L} = \partial_{p_i}\mathbb{E}_{p(z)}[f(z)]$ for the $i$-th dimension. As the random sampling operation in the expectation is not differentiable, Bengio et al. (2013) propose the Straight-Through estimator that approximates the gradient with $\partial_{p_i}\mathcal{L} \approx \mathbb{E}_{p(z)}[\partial_{z_i}f(z)]$. Clearly, the Straight-Through computes a biased approximation to the true gradient.

Next, we quantify the bias in the Straight-Through estimator using the harmonic analysis of $f(x)$. For the quantification of bias we first need the following lemma that connects the REINFORCE gradients with the degree-1 Fourier coefficients. The lemma is an extension of Margulis-Russo (Margulis, 1974; Russo, 1982; O'Donnell, 2014) formula.

**Lemma 1.** *Let $f$ be a Boolean function. Then, the REINFORCE gradient estimates the degree 1 Fourier coefficients $\hat{f}^{(p)}(i)$ of $f$ under $p(z)$,*

$$E_{p(z)}\left[g^i_{REINFORCE}\right] = \partial_{p_i}\mathbb{E}_{p(z)}[f(z)] = \frac{2}{\sigma_i}\hat{f}^{(p)}(i).$$

*Proof.* For compactness and clarity, we provide the proof in the Appendix B.1. □

We introduce new notation. In the following lemma $bias^{(p)}(g^i_{\text{ST}})$ denotes the bias of the $i$-th gradient estimate under the distribution $p$. Also, given the distribution $p(z)$, $p^{i \to 1/2}(z)$ is the distribution for which we set $p(z_i = +1) = p(z_i = -1) = 1/2$ for a given dimension $i$.

**Lemma 2.** *Let $f$ be a Boolean function, $g^i = \partial_{p_i}\mathbb{E}_{p(z)}[f(z)]$ its true gradients and $g^i_{ST} = \mathbb{E}_{p(z)}[\partial_{z_i}f(z)]$ its Straight-Through gradient estimator. The bias in $g^i_{ST}$ is due to a mismatch between the higher-order odd terms in the Taylor expansion of $g^i$ and $g^i_{ST}$ and is equal to*

$$bias^{(p)}\left(g^i_{ST}\right) = bias^{(p^{i \to 1/2})}\left(g^i_{ST}\right) + \mathbb{E}\left[\sum_{k=2j, j>0} kc_k\right]\mu_i \tag{4}$$

*where $c_k$ are the Taylor coefficients for the $i$-th dimension on $f(z)$, that is $z_i$, around 0 and $bias^{(p^{i \to 1/2})} = \mathbb{E}\left[\sum_{k=2j+1, j>0}(k-2)c_k\right]$.*

*Proof.* For compactness and clarity, we provide only a proof sketch here showing the basic steps. These steps are also needed later in the description of the proposed gradient estimate algorithm. For the detailed proof please refer to the Appendix B.2. The proof sketch goes as follows. First, we derive a relation between the Fourier coefficients under the unknown distribution $p(z)$ and under the uniform Bernoulli distribution $p^{i \to 1/2}(z)$. Then, using this relation we derive the Taylor expansions for the true gradient as well as the Straight-Through gradient estimator. Last, to prove the lemma we compare the two Taylor expansions.

**Relation between Fourier coefficients under $p(z)$ and $p^{i \to 1/2}(z)$.** If we expand the function $f$ in terms of its $\phi_S$ basis as in equation 2 and focus on the $i$-th dimension, by Lemma 1 we can show that the REINFORCE gradient is given by

$$\partial_{p_i}\mathbb{E}_{p(z)}[f(z)] = 2\frac{\hat{f}^{(p)}(i)}{\sigma_i} = 2\hat{f}^{(p \to 1/2)}(i) \tag{5}$$

**Taylor expansions of the true and the Straight-Through gradients.** The Taylor expansion of $f(z)$ for $z_i$ around 0 is given by $f(z) = c_0 + c_1 z_i + c_2 z_i^2 + c_3 z_i^3 + c_4 z_i^4 + c_5 z_i^5 + \cdots$, where $c_k k! = \partial_{z_i}^k f(z)|_{z_i=0}$ are the Taylor coefficients. All $c_k$ are a function of $z_j, j \neq i$.

Let's first focus on the true gradient. Since we work with Boolean $\pm 1$ values, we have that $z_i^k = 1$ for even $k$ and $z_i^k = z_i$ for odd $k$. This will influence the even and the odd terms of the Taylor expansions. Specifically, for the Taylor expansion of the true gradient we can show that

$$\partial_{p_i}[\mathbb{E}_{p(z)}[f(z)]] = 2\hat{f}^{(p \to 1/2)}(i) = 2\mathbb{E}_{p^{i \to 1/2}(z)}[c_0 z_i + c_1 z_i^2 + c_i z_i^3 + \cdots] \tag{6}$$
$$= 2\mathbb{E}_{p(z_{\backslash i})}[c_1 + c_3 + c_5 + \cdots] \tag{7}$$

The expression in equation 7 implies that the true gradient with respect to the $p_i$ is the expected sum of the odd Taylor coefficients. Here we note that although the final expression in equation 7 can also be derived by a finite difference method, it does not make explicit, as in equation 31, the dependence on $z_i$ and $\mu_i$ of the term inside the expectation.

Now, let's focus on the Straight-Through gradient. Taking the derivative of the Taylor expansion w.r.t. to $z_i$, we have

$$\partial_{z_i} f(z) = c_1 + 2c_2 z_i + 3c_3 z_i^2 + 4c_4 z_i^3 + 5c_5 z_i^4 + \ldots \tag{8}$$

The Straight-Through gradient is the expectation of equation 8 in the $i$-th dimension, that is

$$\mathbb{E}_{p(z)}[\partial_{z_i} f(z)] = \mathbb{E}_{p(z)}[c_1 + 2c_2 z_i + 3c_3 z_i^2 + 4c_4 z_i^3 + 5c_5 z_i^4 + \cdots] \tag{9}$$
$$= \mathbb{E}_{p(z_{\backslash i})}[c_1 + 3c_3 + 5c_5 + \cdots] + \mathbb{E}_{p(z_{\backslash i})}[2c_2 + 4c_4 + \cdots]\mu_i \tag{10}$$

where $z_i^{2j} = 1, z_i^{2j+1} = z_i$, and $\mathbb{E}_{p(z_i)}[z_i] = \mu_i$.

**Comparing the Taylor expansions.** By comparing the expansion of the Straight-Through gradient in equation 10 and the expansion of the true gradient in equation 7 and given that $bias^{(p)}(g_{\text{ST}}^i) = \mathbb{E}_{p(z)}[\partial_{z_i} f(z)] - \partial_{p_i}\mathbb{E}_{p(z)}[f(z)]$,

$$bias^{(p)}(g_{\text{ST}}^i) = \mathbb{E}_{p(z_{\backslash i})}\left[\sum_{k=2j+1, j>0}(k-2)c_k\right] + \mathbb{E}_{p(z_{\backslash i})}\left[\sum_{k=2j, j>0}k c_k\right]\mu_i. \tag{11}$$

Taking the expectation in equation 9 under $p^{i \to 1/2}$ causes the final term in equation 11 to vanish leaving $bias^{(p^{i \to 1/2})}(g_{\text{ST}}^i) = \mathbb{E}_{p(z_{\backslash i})}\left[\sum_{k=2j+1, j>0}(k-2)c_k\right]$. Combining this expression with equation 9 gives the final expression (equation 4) from the lemma. $\qquad\square$

## 4 LOW-BIAS GRADIENT ESTIMATES

### 4.1 FOUST GRADIENT ESTIMATION ALGORITHM

Inspired by the Harmonic Analysis of the Straight-Through gradient estimates, we present a gradient estimate algorithm for deep Boolean latent models, *FouST*, for Fourier Straight-Through estimator. The algorithm relies on three bias reduction steps on the Straight-Through, lines 2, 3, 5 in Algorithm 1.

### 4.2 LOWERING BIAS BY IMPORTANCE SAMPLING CORRECTION

As detailed earlier, the bias in the Straight-Through estimator is the sum of the bias under the uniform Bernoulli distribution plus extra bias due to non-zero expectation terms in higher-order harmonics when sampling from $p(z)$. Sampling from $p^{i \to 1/2}$ instead of $p(z)$ would decrease the total bias from the form in equation 11 by setting the final term to 0.

As a first bias reduction step, therefore, we do importance sampling. Specifically, after getting samples from $p(z)$ and computing the gradients $\partial_{z_i} f(z)$ with the Straight-Through, we estimate the expectation under $p^{i \to 1/2}$ as

$$\mathbb{E}_{p^{i \to 1/2}(z)}[\partial_{z_i} f(z)] = \mathbb{E}_{p(z)}\Big[\frac{1}{2p(z_i)}\partial_{z_i} f(z)\Big] \qquad (p^{i \to 1/2}(z_i) = 1/2) \qquad (12)$$

Interestingly, Gregor et al. (2013) arrive at equation 12 in the context of unbiased control variates for quadratic functions.

### 4.3 LOWERING BIAS BY MATCHING UNIFORM DISTRIBUTION MOMENTS

Lemma 2 shows that part of the bias in the Straight-Through estimator is due to the presence of extra factors in the Taylor coefficients. We can reduce the effect of these factors by taking advantage of the moments of the uniform distribution. Recalling that $\int_0^b \frac{1}{b} u^k du = \frac{b^k}{k+1}$, we can attempt to correct the coefficients in equation 9, which for $z^k$ have the form $(k+1)c_k$, with the same extra factor of $k+1$ that appears in the denominator of the $k$th moment. This suggests that we can sample from an auxiliary variable $u$ and then use the auxiliary variable $u$ with $f$ instead of $z$ and exploit the higher moments of the uniform distribution to reduce bias.

For brevity, we illustrate the method with a case study of a two-dimensional $z$, and a bivariate $f(z_1, z_2)$.

As in Lemma 2, the partial true gradient of $f(z_1, z_2)$ w.r.t. the first distribution parameter $p_1$ equals to

$$\partial_{p_1} \mathbb{E}_p(z) = \sum_j c_{1,j} \mathbb{E}_{p(z)}[z_2^j] + \sum_j c_{3,j} \mathbb{E}_{p(z)}[z_2^j] + \sum_j c_{5,j} \mathbb{E}_{p(z)}[z_2^j] + \cdots \qquad (13)$$

**"Bernoulli splitting uniform" trick.** Assume an auxiliary variable $u = (u_1, u_2)$, which we choose as follows. First, we sample $z = (z_1, z_2)$ from the uniform Bernoulli distribution $p^{1 \to 1/2}$ with ($i$ set to 1). Then we take a uniform sample $(u_1, u_2)$ with $u_i$ sampled from either $[0, 1]$ for $z_i = +1$ or from $[-1, 0]$ if $z_i = -1$.

The expectation of the gradient under such random sampling is

$$\mathbb{E}_{z,u}[\partial_{z_1} f(u_1, u_2)] = \sum_j \frac{c_{1,j}}{j+1} \mathbb{E}_z[z_2^j] + \sum_j \frac{c_{3,j}}{j+1} \mathbb{E}_z[z_2^j] + \cdots \qquad (14)$$

Further detail is in Appendix C.1. We compare equation 14 with equation 13. In equation 14 we observe that the pure terms in $z_1$, namely terms with $j = 0$, always match those of the true gradient in equation 13. For $j > 0$ we obtain mixed terms with coefficients that do not match those of the true gradient in equation 13. Due to the $\frac{1}{j+1}$ factor, for small $j$ the mixed-degree terms are closer to the original ones in equation 13. For functions with small mixed degree terms, this can lead to bias reduction, at the cost of an increased variance because of sampling an auxiliary variable. In practice, to manage this bias-variance trade-off and to deal with functions that have greater dependence on mixed degree terms, we use smaller intervals for the random samples as in Algorithm 1.

To summarize, for appropriate functions, the "Bernoulli splitting uniform" relies on the *continuous* variable $u$ conditioned on the binary sample to reduce the bias. However, it is important to emphasize that $u$ is only an auxiliary variable; the actual latent variable $z$ is *always binary*. Thus, the "Bernoulli splitting uniform" trick *does not lead to a relaxation* of the sort used by Gumbel-Softmax (Jang et al., 2016), where there are no hard samples. Lastly we note that for a univariate $f$ the "Bernoulli splitting uniform" trick leads to an unbiased estimator with an increased variance.

### 4.4 LOWERING BIAS BY REPRESENTATION SCALING

The Fourier basis does not depend on the particular input representation and any two-valued set, say $\{-t, t\}$ can be used as the Boolean representation. The choice of a representation, however, does affect the bias as we show next. As a concrete example, we let our input representation be $z_i \in \{-1/2, 1/2\}^n$, where $p_i = p(z_i = +1/2)$. While we can change the input representation like that, in general the Fourier coefficients in equation 3 will be different than for the $\{-1, +1\}$ representation. We give the final forms of the gradients here. Details are given in Appendix C.2.

Under the $p^{i \to 1/2}$ distribution the degree-1 Fourier coefficients are:

$$\hat{f}^{(p^{i \to 1/2})}(i) = \mathbb{E}_{p^{i \to 1/2}(z)} \left[ \frac{c_1}{2} + \frac{c_3}{2^3} + \frac{c_5}{2^5} + \cdots \right] \tag{15}$$

Note that compared to equation 7, in equation 15 we still get the odd terms $c_1, c_3$ albeit decayed by inverse powers of 2. Following the same process for the Straight-Through gradient as in equation 10, we have that

$$\frac{1}{2} \mathbb{E}_{p^{i \to 1/2}} [\partial_{z_i} f(z)] = \mathbb{E}_{p^{i \to 1/2}} \left[ \frac{c_1}{2} + \frac{3c_3}{2^3} + \frac{5c_5}{2^5} + \ldots \right] \tag{16}$$

While this is still biased, compared to equation 7 the bias is reduced by damping higher order terms by inverse powers of 2.

### 4.5 ALGORITHMIC COMPLEXITY

The algorithm, described in algorithm 1, is a Straight-Through gradient estimator with the bias reduction steps described above, where a single sample is used to estimate the gradient. We emphasize that the algorithm uses a single sample and a single evaluation of the decoder per example and latent vector sample. Thus, the algorithm has the same complexity as that of the original Straight-Through estimator.

## 5 RELATED WORK

Monte Carlo gradient estimators for training models with stochastic variables can be biased or unbiased. Perhaps the best known example of an unbiased gradient estimator is the REINFORCE algorithm (Williams, 1992). Unfortunately, REINFORCE gives gradients of high variance. For continuous stochastic variables Kingma & Welling (2013) propose the reparameterization trick, which transforms the random variable into a function of deterministic ones perturbed by a fixed noise source, yielding much lower variance gradient estimates. For discrete stochastic variables, REINFORCE is augmented with *control variates* for variance reduction. A number of control variate schemes have been proposed: NVIL (Mnih & Gregor, 2014) subtracts two baselines (one constant and one input-dependent) from the objective to reduce variance. MuProp (Gu et al., 2015) uses the first-order Taylor approximation of the function as a baseline. REBAR (Tucker et al., 2017) uses the Gumbel-Softmax trick to form a control variate for unbiased gradient estimates. RELAX (Grathwohl et al., 2017) generalizes REBAR to include an auxiliary network in the gradient expression and uses continuous relaxations and the reparameterization trick to give unbiased gradients.

Regarding biased estimators, a simple choice is the Straight-Through estimator (Bengio et al., 2013) which uses the gradient relative to the sample as that relative to the probability parameter. Another recent approach is to use continuous relaxations of discrete random variables so that the reparameterization trick becomes applicable. The most common example of this being the Gumbel-Softmax estimator (Maddison et al., 2016; Jang et al., 2016). Although this is a continuous relaxation, it has been used to define the Gumbel Straight-Through estimator with hard samples. This uses

$\arg\max$ in the forward pass and the Gumbel-Softmax gradient is used as an approximation during in the backward pass. DARN (Gregor et al., 2013), like MuProp, also uses the first-order Taylor expansion as a baseline but does not add the analytical expectation, making the estimator biased for non-quadratic functions.

In this work we focus on biased Straight-Through gradient estimators. Specifically, we analyse how to reduce bias via Fourier expansions of Boolean functions. The Fourier expansion itself is widely used in computational learning theory with applications to learning low-degree functions (Kushilevitz & Mansour, 1993), decision trees (Mansour, 1994), constant-depth circuits (Linial et al., 1993) and juntas (Mossel et al., 2003). To the best of our knowledge we are the first to explore Fourier expansions for bias reduction of biased stochastic gradient estimators.

## 6 EXPERIMENTS

**Experimental Setup.** We first validate FouST on a toy setup, where we already know the analytic expression of $f(z)$. Next we validate FouST by training generative models using the variational autoencoder framework of Kingma & Welling (2013). We optimize the single sample variational lower bound (ELBO) of the log-likelihood. We train variational autoencoders *exclusively* with Boolean latent variables on OMNIGLOT, CIFAR10, mini-ImageNet (Vinyals et al., 2016) and MNIST (Appendix D.1). We train all models using a regular GPU with stochastic gradient descent with a momentum of 0.9 and a batch size of 128. We compare against Straight-Through, Gumbel-Softmax, and DARN, although on more complex models some estimators diverge. The results were consistent over multiple runs. Details regarding the architectures and hyperparameters used are in Appendix E. *Upon acceptance we will open source all code, models, data and experiments.*

### 6.1 BIASED ESTIMATORS ON A TOY PROBLEM

To validate the excessive bias in the Straight-Through estimator, as well as to explore the benefits of FouST, we first explore a toy problem. Similar to Tucker et al. (2017), we minimize $\mathbb{E}_{p(z)}[(z-t)^2]$, where $t \in (0,1)$ is a continuous target value and $z$ is a sample from the Bernoulli distribution $p(z)$. The optimum is obtained for $p(z = +1) \in \{0,1\}$. Figure 1, shows a case with $t = 0.45$, where the minimizing solution is $p(z = +1) = 0$. We observe that unlike the Straight-Through estimator, FouST converges to the minimizing deterministic solution (lower is better).

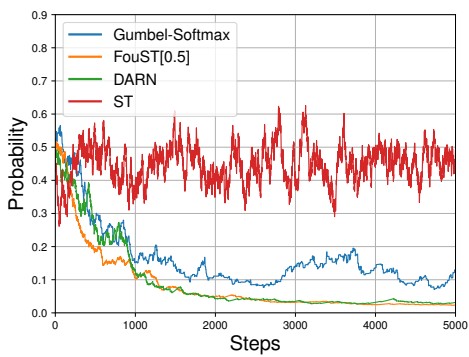

Figure 1: Biased estimators on a toy problem

### 6.2 FOUST VS. STATE-OF-THE-ART GRADIENT ESTIMATORS

**Training Stochastic MLPs.** We train MLPs with one and two stochastic layers on OMNIGLOT, following the non-linear architecture of Tucker et al. (2017). Each stochastic Boolean layer is preceded by two deterministic layers of 200 $\tanh$ units.

All hyperparameters remain fixed throughout the training. All estimators use *one sample per example and a single decoder evaluation*. We present results in Fig. 2. FouST outperforms other biased gradient estimators in both datasets and architectures. FouST is clearly better than the Straight-Through estimator. Despite the complicated nature of the optimized neural network function $f(z)$ the bias reduction appears fruitful. With one or two stochastic layers we can also use the unbiased REBAR. REBAR is not directly comparable to the estimators we study, since it uses multiple decoder evaluations and for models with multiple stochastic layers, multiple passes through later layers. Nevertheless, as shown in appendix D.1 for MNIST, with two stochastic layers REBAR reaches a worse test ELBO of -94.43 v. -91.94 for FouST. A possibility of worse test than training ELBOs for REBAR was also suggested in the original work (Tucker et al., 2017).

**Training Stochastic ResNets.** We further validate FouST in a setting where the encoder and decoder are stochastic ResNets, S-ResNets, which are standard ResNets with stochastic layers inserted between ResNet blocks. Similar to MLPs, FouST outperforms other biased estimators in this

setting on CIFAR-10 (left in Figure 3). Note that despite the hyperparameter sweep, we were unable to train S-ResNet's with Gumbel-Softmax. So we compare against DARN and Straight-Through only. With an S-ResNet with 12 ResNet blocks and 4 stochatic layers FouST yields a score of 5.08 bits per dimension (bpd). This is comparable to the 5.14 bpd with the categorical VIMCO-trained model (Mnih & Rezende, 2016) obtained with 50 samples per example, as reported by van den Oord et al. (2017).

In the plots, we observe sharp spikes or slower curving. We hypothesize these are due, respectively, to stochasticity and bias, and are corrected to some degree along the way.

**Efficiency.** We compare the efficiency of different estimators in Tab. 1. Like other biased estimators, FouST requires a single sample for estimating the gradients and has similar wallclock times. On MNIST, the unbiased REBAR is 15x and 40x slower than the biased estimators for two and five stochastic layer MLP's respectively.

Table 1: Wall clock times for various gradient estimators on MNIST.

| Method | #Eval. | Walltime in sec./Epoch | |
| --- | --- | --- | --- |
| | | 2 layers | 5 layers |
| REBAR | 3 | 45.3 | 205.15 |
| ST | 1 | 2.86 | 4.96 |
| Gumbel | 1 | 3.27 | 6.14 |
| DARN | 1 | 2.96 | 5.36 |
| FouST | 1 | 3.1 | 5.67 |

From the above experiments we conclude that FouST allows for efficient and effective training of fully connected and convolutional neural networks with Boolean stochastic variables.

## 6.3 ADDING STOCHASTIC DEPTH AND WIDTH

Last, we evaluate FouST on more complex neural networks with deeper and wider stochastic layers. We perform experiments with convolutional architectures on the larger scale and more realistic mini-ImageNet (Vinyals et al., 2016). As the scope of this work is not architecture search, we present two architectures inspired from residual networks (He et al., 2016) of varying stochastic depth and width. The first one is a wide S-ResNet, S-ResNet-40-2-800, and has 40 deterministic (with encoder and decoder combined), 2 stochastic layers, and 800 channels for the last stochastic layer. The second, S-ResNet-80-11-256, is very deep with 80 deterministic and 11 stochastic layers, and a last stochastic layer with 256 channels. Architecture details are given in Appendix E.2. In this setup, training with existing unbiased estimators is intractable.

We present results in Fig. 3. We compare against DARN, since we were unable to train the models with Gumbel-Softmax. Incomplete lines indicate failure.

We observe that FouST is able to achieve better training ELBO's in both cases. We conclude that FouST allows for scaling up the complexity of stochastic neural networks in terms of stochastic depth and width.

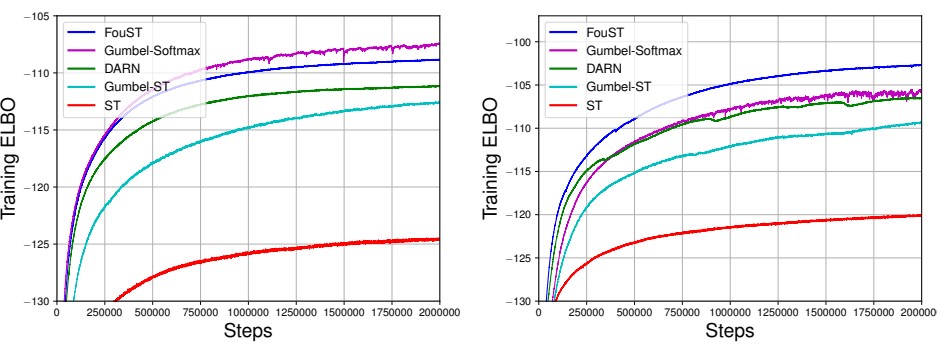

Figure 2: Training ELBO for the one (left) and two (right) stochastic layer nonlinear models on OMNIGLOT

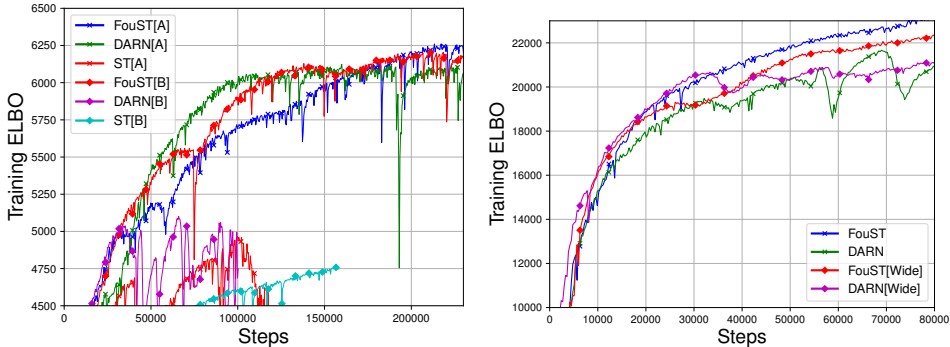

Figure 3: Training ELBO on CIFAR-10 (left) and mini-ImageNet (right)

## 7 CONCLUSION

In this work we introduce the framework of harmonic analysis for Boolean functions. We then use the harmonic analysis to derive the source of bias in the Straight-Through estimator for Boolean random variables. Based on the analysis we propose FouST, which is a novel gradient estimate algorithm. We use FouST to train neural networks with Boolean random variables in stochastic layers. FouST outperforms state-of-the-art biased gradient estimators, while maintaining efficiency that is orders of magnitude higher than unbiased estimators. Importantly, to the best of our knowledge FouST is the first gradient estimate algorithm, biased or unbiased, to be able to train very deep and wide neural networks with Boolean random variables. In our experiments FouST successfully trained networks with up to 80 deterministic and 11 stochastic layers. We conclude that the Harmonic Analysis framework for Boolean function is a useful methodological tool for analyzing Boolean neural networks. Also, we conclude that FouST is a practical gradient estimate algorithm that can train complex stochastic neural networks with multiple layers of Boolean random variables.

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

## APPENDICES

# A   FURTHER DETAILS FROM THE HARMONIC ANALYSIS FRAMEWORK FOR BOOLEAN FUNCTIONS

## A.1   DISCRETE DERIVATIVE

For a Boolean function $f$ the discrete derivative on the $i$-th latent dimension with a basis function $\phi_i$ is defined as

$$D_{\phi_i} f(z) = \sigma_i \frac{f(z_1, ..., z_i = +1, ..., z_n) - f(z_1, ..., z_i = -1, ..., z_n)}{2}. \tag{17}$$

The Fourier expansion of the discrete derivative equals $D_{\phi_i} f(z) = \sum_{S \ni i} \hat{f}^{(p)}(S) \phi_{S \setminus i}(z)$, The Fourier expansion of the discrete derivative is derived by equation 2: *(i)* all bases that do not contain the $i$-th dimension are constant to $\phi_i$ and thus set to zero, while *(ii)* for the rest of the terms $\frac{\partial \phi_S}{d\phi_i} = \phi_{S \setminus i}$ from the definition of basis functions $\phi_S(z)$. In the following we differentiate partial derivatives on continuous functions noted with $\partial \cdot$ from discrete derivatives on Boolean functions noted with $D \cdot$. The $i$-th discrete derivative is independent of $z_i$ both in the above definitions.

## B  HARMONIC ANALYSIS OF EXISTING GRADIENT ESTIMATES

### B.1  PROOF OF LEMMA 1

*Proof.* We follow O'Donnell (2014, §8.4). In this proof we work with two representations of the Boolean function $f$. The first is the Fourier expansion of $f$ under the uniform Bernoulli distribution. This is also the representation obtained by expressing $f$ as a polynomial in $z_i$. Since the domain of the function $f$ is the Boolean cube, the polynomial representation is multilinear. That is $f(z) = \sum_{S \subseteq [n]} \hat{f}(S) \prod_{j \in S} z_j$. To avoid confusion and to differentiate the representation from the Boolean function, we use $f^{(u)}(z)$ to denote this representation in the following. Note that since this representation is a polynomial it is defined over any input in $\mathbb{R}^n$. In particular,

$$\mathbb{E}[f(z)] = \mathbb{E}[\sum_{S \subseteq [n]} \hat{f}(S) \prod_{j \in S} z_j] = \sum_{S \subseteq [n]} \hat{f}(S) \prod_{j \in S} \mathbb{E}[z_j] = f^{(u)}(\mu_1, \ldots, \mu_n)$$

The second representation we use is the Fourier expansion of the Boolean function $f$ under $p(x)$. We denote this by $f^{(p)}$.

The following relation follows from the fact that when working with the Fourier representation, $f(z)$ is multilinear, $\mathbb{E}_{p(z)}[z_i] = \mu_i$ and the linearity of expectation.

$$\mathbb{E}_{p(z)}[f^{(p)}(z_1, \ldots, z_n)] = \mathbb{E}_{p(z)}[f(z_1, \ldots, z_n)] = f^{(u)}(\mu_1, \ldots, \mu_n). \tag{18}$$

As the partial derivative of $f^{(u)}$ w.r.t. $\mu_i$ is equivalent to discrete derivative of $f^{(u)}$ w.r.t. $z_i$, $\partial_{\mu_i} f^{(u)}(\mu) = D_{z_i} f^{(u)}(\mu)$, and keeping in mind that $\phi_i = (z_i - \mu_i)/\sigma_i$, we have that

$$D_{z_i} f^{(u)}(\mu) = \mathbb{E}_{p(z)}[D_{z_i} f^{(p)}(z_1, \ldots, z_n)] \qquad \text{(from } equation \text{ 18)} \tag{19}$$

$$= \frac{1}{\sigma_i} \mathbb{E}_{p(z)}[D_{\phi_i} f^{(p)}(z_1, \ldots, z_n)] \qquad \text{(set } \phi_i = (z_i - \mu_i)/\sigma_i, \text{ then chain rule)} \tag{20}$$

$$= \frac{1}{\sigma_i} \hat{f}^{(p)}(i) \tag{21}$$

We then note that the discrete derivative of $f$ w.r.t. $z_i$, $D_{z_i} f^{(u)}(\mu)$, from the left hand side of equation 19, is equivalent to the partial derivative of $f$ w.r.t. $\mu_i$, $\partial_{\mu_i} f^{(u)}(\mu)$.

$$\frac{1}{\sigma_i} \hat{f}^{(p)}(i) = D_{z_i} f^{(u)}(\mu) = \partial_{\mu_i} f^{(u)}(\mu) \tag{22}$$

$$= \frac{1}{2} \partial_{p_i} f^{(u)}(\mu) \qquad \text{(set } \mu_i = 2p_i - 1, \text{ then chain rule)} \tag{23}$$

$$= \frac{1}{2} \partial_{p_i} \mathbb{E}_{p(z)}[f^{(p)}(z)] \qquad \text{(from } equation \text{ 18)} \tag{24}$$

We complete the proof by noting that the right hand side in equation 24 is $\frac{1}{2}$ times the REINFORCE gradient.  □

### B.2  ANALYSIS OF STRAIGHT-THROUGH

The detailed proof of Lemma 2 is as follows.

*Proof.* We first derive the Taylor expansions for the true gradient as well as the Straight-Through gradient estimator. Then, we prove the lemma by comparing the two Taylor expansions.

By expanding the function $f$ in terms of its $\phi_S$ basis as in equation 2 and focusing on the $i$-th dimension, we have that

$$\hat{f}^{(p)}(i)\phi_i = \hat{f}^{(p)}(i)\frac{z_i - \mu_i}{\sigma_i} = \hat{f}^{(p)}(i)\frac{z_i}{\sigma_i} - \hat{f}^{(p)}(i)\frac{\mu_i}{\sigma_i} \tag{25}$$

The first term, $\hat{f}^{(p)}(i)\frac{z_i}{\sigma_i}$, is the term corresponding to $\{i\}$ in the Fourier expansion of $f$ under $p^{i \to 1/2}(z)$. That is

$$\hat{f}^{(p \to 1/2)}(i) = \frac{1}{\sigma_i} \hat{f}^{(p)}(i) \tag{26}$$

This follows from the fact that when moving from $p(z)$ to $p^{i \to 1/2}(z)$, *(i)* we have that $\phi_i = z_i$, and *(ii)* no other term under the $p(z)$ expansion contributes to the $z_i$ term under the $p^{i \to 1/2}(z)$ expansion.

As a consequence of Lemma 1 the REINFORCE gradient for the $i$-th dimension is given by

$$\partial_{p_i} \mathbb{E}_{p(z)}[f(z)] = 2\frac{\hat{f}^{(p)}(i)}{\sigma_i} = 2\hat{f}^{(p\rightarrow 1/2)}(i) \tag{27}$$

Next, we will derive the Taylor expansions of the true and the Straight-Through gradients. The Taylor expansion of $f(z)$ for $z_i$ around 0 is

$$f(z) = c_0 + c_1 z_i + c_2 z_i^2 + c_3 z_i^3 + c_4 z_i^4 + c_5 z_i^5 + ..., \tag{28}$$

where $c_k k! = \partial_{z_i}^k f(z)|_{z_i=0}$ are the Taylor coefficients. All $c_k$ are a function of $z_j, j \neq i$.

Let's first focus on the true gradient. Since we work with binary $\pm 1$ values, we have that $z_i^k = 1$ for even $k$ and $z_i^k = z_i$ for odd $k$. This will influence the even and the odd terms of the Taylor expansions. Specifically, for the Taylor expansion of the true gradient we have from equation 27 and equation 3 that

$$\partial_{p_i}[\mathbb{E}_{p(z)}[f(z)]] = 2\hat{f}^{(p\rightarrow 1/2)}(i) = 2\mathbb{E}_{p^{i\rightarrow 1/2}(z)}[(c_0 + c_1 z_i + c_2 z_i^2 + \cdots)z_i] \tag{29}$$

$$= 2\mathbb{E}_{p^{i\rightarrow 1/2}(z)}[c_0 z_i + c_1 z_i^2 + c_i z_i^3 + \cdots] \tag{30}$$

$$= 2\mathbb{E}_{p^{i\rightarrow 1/2}(z)}[c_0 z_i + c_1 + c_2 z_i + c_3 + \cdots] \quad (z_i^{2j} = 1, z_i^{2j+1} = z_i, \mathbb{E}_{p^{i\rightarrow 1/2}(z)}[z_i] = 0) \tag{31}$$

$$= 2\mathbb{E}_{p(z_{\backslash i})}[c_1 + c_3 + c_5 + \cdots] \tag{32}$$

The expression in equation 32 implies that the true gradient with respect to the $p_i$ is the expected sum of the odd Taylor coefficients. Here we note that the although final expression in equation 32 can also be derived by a finite difference method, it does not make explicit, as in equation 31, the dependence on $z_i$ and $\mu_i$ of the term inside the expectation.

Now, let's focus on the Straight-Through gradient. Taking the derivative of the Taylor expansion in equation 28 w.r.t. to $z_i$, we have

$$\partial_{z_i} f(z) = c_1 + 2c_2 z_i + 3c_3 z_i^2 + 4c_4 z_i^3 + 5c_5 z_i^4 + ... \tag{33}$$

The Straight-Through gradient is the expectation of equation 33 in the $i$-th dimension, that is

$$\mathbb{E}_{p(z)}[\partial_{z_i} f(z)] = \mathbb{E}_{p(z)}[c_1 + 2c_2 z_i + 3c_3 z_i^2 + 4c_4 z_i^3 + 5c_5 z_i^4 + \cdots] \tag{34}$$

$$= \mathbb{E}_{p(z_{\backslash i})}[c_1 + 3c_3 + 5c_5 + \cdots] + \mathbb{E}_{p(z_{\backslash i})}[2c_2 + 4c_4 + \cdots]\mu_i \quad (z_i^{2j} = 1, z_i^{2j+1} = z_i, \mathbb{E}_{p(z_i)}[z_i] = \mu_i) \tag{35}$$

By comparing the expansion of the Straight-Through gradient in equation 35 and the expansion of the true gradient in equation 32,

$$bias^{(p)}(g_{ST}^i) = \mathbb{E}_{p(z)}[\partial_{z_i} f(z)] - \partial_{p_i}\mathbb{E}_{p(z)}[f(z)] \tag{36}$$

$$= \mathbb{E}_{p(z_{\backslash i})}\left[\sum_{k=2j+1, j>0}(k-2)c_k\right] + \mathbb{E}_{p(z_{\backslash i})}\left[\sum_{k=2j, j>0}kc_k\right]\mu_i. \tag{37}$$

Taking the expectation in equation 34 under $p^{i\rightarrow 1/2}$ causes the final term in equation 37 to vanish leaving

$$bias^{(p^{i\rightarrow 1/2})}(g_{ST}^i) = \mathbb{E}_{p(z_{\backslash i})}\left[\sum_{k=2j+1, j>0}(k-2)c_k\right]. \tag{38}$$

$\square$

# C  LOW-BIAS GRADIENT ESTIMATES

## C.1  LOWERING BIAS BY MATCHING UNIFORM DISTRIBUTION MOMENTS

We describe the case of a bivariate function in detail.

For brevity, we focus on a case study of a two-dimensional $z$, and a bivariate $f(z_1, z_2)$ with the bivariate Taylor expansion $f(z_1, z_2) = \sum_{i,j} c_{i,j} z_1^i z_2^j$.

As in Lemma 2, the partial true gradient of $f(z_1, z_2)$ w.r.t. the first distribution parameter $p_1$ equals to

$$\partial_{p_1}\mathbb{E}_p(z) = \sum_j c_{1,j}\mathbb{E}_{p(z)}[z_2^j] + \sum_j c_{3,j}\mathbb{E}_{p(z)}[z_2^j] + \sum_j c_{5,j}\mathbb{E}_{p(z)}[z_2^j] + ... \tag{39}$$

Further, the Taylor expansion of $\partial_{p_1} f(z_1, z_2)$ is $\partial_{z_1} f(z_1, z_2) = \sum_j c_{1,j} z_2^j + 2\sum_j c_{2,j} z_2^j x_1 + 3\sum_j c_{3,j} z_2^j z_1^2 + \cdots$

**"Bernoulli splitting uniform" trick.** Assume an auxiliary variable $u = (u_1, u_2)$, which we choose as follows. First, we sample $z = (z_1, z_2)$ from the uniform Bernoulli distribution $p^{1 \to 1/2}$. Then we take a uniform sample $(u_1, u_2)$ with $u_i$ sampled from either $[0, 1]$ for $z_i = +1$ or from $[-1, 0]$ if $z_i = -1$. At this point it is important to note that the moments of the uniform distribution in $[a, b]$ are $\int_a^b \frac{1}{b-a} u^k = \frac{1}{b-a} \frac{b^{k+1} - a^{k+1}}{k+1}$, which simplifies to $b/2, b^2/3, b^3/4, \ldots$ for $a = 0$, and where we think of $b$ as a binary sample i.e., $b \in \{-1, 1\}$. The expectation of the gradient under such random sampling is

$$\mathbb{E}_{z,u}[\partial_{z_1} f(u_1, u_2)] = \sum_j c_{1,j} \mathbb{E}_{z,u}[u_2^j] + 2 \sum_j c_{2,j} \mathbb{E}_{z,u}[u_2^j] \mathbb{E}_{z,u}[u_1] + 3 \sum_j c_{3,j} \mathbb{E}_{z,u}[u_2^j] \mathbb{E}_{z,u}[u_1^2] + \cdots$$

$$= \sum_j \frac{c_{1,j}}{j+1} \mathbb{E}_z[z_2^j] + \sum_j \frac{c_{3,j}}{j+1} \mathbb{E}_z[z_2^j] + \cdots \tag{40}$$

We then compare equation 40 with equation 39. In equation 40 we observe that the pure terms in $z_1$, namely terms with $j = 0$, always match those of the true gradient in equation 39. For $j > 0$ we obtain mixed terms with coefficients that do not match those of the true gradient in equation 39. However, the partial gradient obtained with the auxiliary variables in equation 40 has coefficients following a decaying trend due to the $\frac{1}{j+1}$. For small $j$, that is, the mixed-degree terms are closer to the original ones in equation 39. For functions with smaller mixed degree terms this leads to bias reduction, at the cost of an increased variance due to additional sampling. In practice many functions would have greater dependence on mixed degree terms. For such functions and to manage the bias-variance trade-off we choose smaller intervals for the uniform samples, that is $a \to b$.

## C.2 LOWERING BIAS BY REPRESENTATION SCALING

The Fourier basis does not depend on the particular input representation and any two-valued set, say $\{-t, t\}$ can be used as the Boolean representation. The choice of a representation, however, does affect the bias as we show next. As a concrete example, we let our input representation be $z_i \in \{-1/2, 1/2\}^n$, where $p_i = p(z_i = +1/2)$. While we can change the input representation like that, in general the Fourier coefficients in equation 3 will be different than for the $\{-1, +1\}$ representation. Letting $h(z_i) = 2z_i \in \{-1, 1\}$, the functions $\phi_i$ are now given as $\phi_i = \frac{h(z_i) - \mu_i}{\sigma_i}$.

Next, we write the Taylor series of $f$ in terms of $h(z_i)$,

$$f(z) = c_0 + c_1 z_i + c_2 z_i^2 + c_3 z_i^3 + c_4 z_i^4 + c_5 z_i^5 + \cdots \tag{41}$$

$$= c_0 + \frac{c_1}{2} h(z_i) + \frac{c_2}{2^2} h(z_i)^2 + \frac{c_3}{2^3} h(z_i)^3 + \frac{c_4}{2^4} h(z_i)^4 + \frac{c_5}{2^5} h(z_i)^5 + \cdots \tag{42}$$

Under the $p^{i \to 1/2}$ distribution, we still have that $\mathbb{E}_{p \to 1/2}[h(z_i)] = 0$ and the degree-1 Fourier coefficients are:

$$\hat{f}^{(p \to 1/2)}(i) = \mathbb{E}_{p^{i \to 1/2}(z)}[f(z)h(z_i)] = \mathbb{E}_{p^{i \to 1/2}(z)} \left[ \frac{c_1}{2} + \frac{c_3}{2^3} + \frac{c_5}{2^5} + \cdots \right] \tag{43}$$

Note that compared to equation 7, in equation 43 we still get the odd terms $c_1, c_3$ albeit decayed by inverse powers of 2. Following the same process like for equation 10, we have that

$$\frac{1}{2} \mathbb{E}_{p^{i \to 1/2}}[\partial_{z_i} f(z)] = \mathbb{E}_{p^{i \to 1/2}} \left[ \frac{c_1}{2} + \frac{3c_3}{2^3} + \frac{5c_5}{2^5} + \cdots \right] \tag{44}$$

# D EXTRA EXPERIMENTS

## D.1 EXPERIMENTS ON MNIST

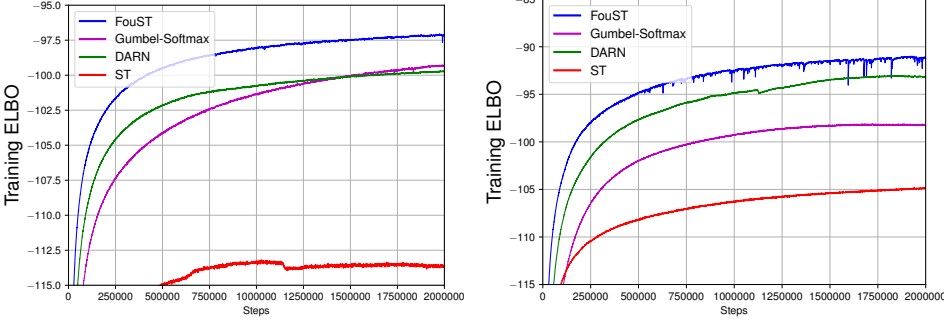

Figure 4: Training ELBO for the one (left) and two (right) stochastic layer nonlinear models on MNIST

Table 2: Test set performance with increasing stochastic depth on MNIST.

| Method | Stochastic Layers | Hidden Units per Layer | Test ELBO |
|--------|-------------------|------------------------|-----------|
| Rebar  | 2  | 200 | -94.43 |
| FouST  | 2  | 200 | -91.95 |
| FouST  | 3  | 200 | -89.11 |
| FouST  | 8  | 500 | -87.31 |
| FouST  | 20 | 500 | -87.86 |

## D.2 ABLATION EXPERIMENTS

To further judge the effect of our proposed modifications to Straight-Through, we performed ablation experiments where we separately applied scaling and noise to the importance-corrected Straight-Through. These experiments were performed on the single stochastic layer MNIST and OMNIGLOT models.

The results of the ablation experiments are shown in figure 5. From the figure it can be seen that scaling alone improves optimization in both cases and noise alone helps in the case of MNIST. Noise alone results in a worse ELBO in the case of OMNIGLOT, but gives an improvement when combined with scaling. From these results we conclude that the proposed modifications are effective.

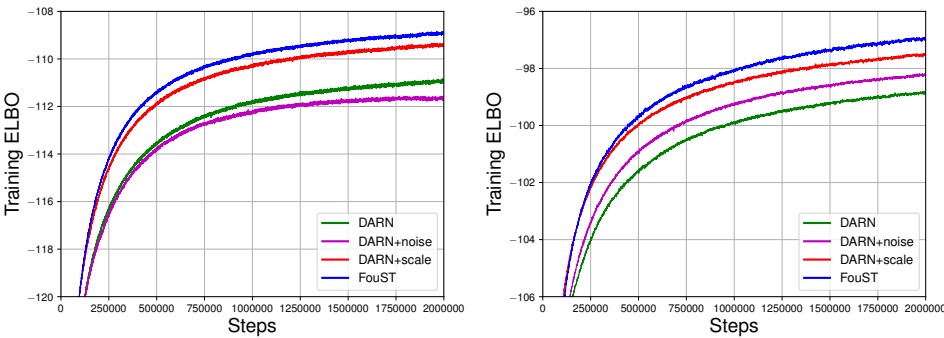

Figure 5: Ablations for the one stochastic layer nonlinear model on OMNIGLOT (left) and MNIST (right)

## E ARCHITECTURES USED IN THE EXPERIMENTS

### E.1 ARCHITECTURES FOR MNIST AND OMNIGLOT

The encoder and decoder networks in this case are MLP's with one or more stochastic layers. Each stochastic layer is preceded by 2 deterministic layers with a tanh activation function.

We chose learning rates from $\{1 \times 10^{-4}, 2 \times 10^{-4}, 4 \times 10^{-4}, 6 \times 10^{-4}\}$, Gumbel-Softmax temperatures from $\{0.1, 0.5\}$, and noise interval length for FouST from $\{0.1, 0.2\}$.

### E.2 ARCHITECTURES FOR CIFAR-10 AND MINI-IMAGENET

For these dataset we use a stochastic variant or ResNets (He et al., 2016). Each network is composed of *stacks* of *layers*. Each layer has *(i)* one regular residual block as in He et al. (2016), *(ii)* followed by at most one stochastic layer, except for the CIFAR architecture B in figure 3 where we used two stochastic layers in the last layer. The stacks are followed by a final stochastic layer in the encoder. We do downsampling at most once per stack. We used two layers per stack.

For CIFAR we downsample twice so that the last stochastic layer has feature maps of size 8x8. We chose learning rate from $\{9 \times 10^{-7}, 1 \times 10^{-6}, 2 \times 10^{-6}, 4 \times 10^{-6}\}$, the FouST scaling parameter from $\{0.5, 0.8, 0.9\}$, and the uniform interval was scaled by a factor from $\{0.01, 0.05, 0.1\}$

For mini-ImageNet we downsample thrice. We chose the learning rate from $\{2 \times 10^{-7}, 3 \times 10^{-7}, 4 \times 10^{-7}, 5 \times 10^{-7}\}$.

For the decoder the structure of the encoder is reversed and convolutions are replaced by transposed convolutions. The output model is Gaussian with learned mean and variance.

