# OpenReview forum: "Low Bias Gradient Estimates for Very Deep Boolean Stochastic Networks"
_ICLR.cc/2020/Conference — Reject_

### Official Review · AnonReviewer2 · 2019-10-23
**Official Blind Review #2**

**Rating:** 6

**Review:**

***Score updated to weak accept after the rebuttal.***

Straight-Through is a popular, yet not theoretically well-understood, biased gradient estimator for Bernoulli random variables. The low variance of this estimator makes it a highly useful tool for training large-scale models with binary latents. However, the bias of this estimator may cause divergence in training, which is a significant practical issue. The paper develops a Fourier analysis of the Straight-Through estimator and provides an expression for the bias of the estimator in terms of the Fourier coefficients of the considered function. Motivated by this expression, the paper proposes two modifications of Straight-Through which may reduce the bias of the estimator, at the cost of the variance. The experimental results show advantage of this improved estimator over Gumbel-Softmax and DARN estimator.

While I really like the premise of the paper, I feel that it needs a significant amount of additional work. The text is currently fairly hard to read. The theoretical part of the paper does not quantify the variance of the estimator. The experiments are a bit unfinished and do not include ablations of the proposed modifications of Straight-Through. Most importantly, I think that in the current form the theoretical and the empirical parts of the papers are not well-connected. Because of this, I believe that the paper should currently be rejected, but I encourage the authors to continue this line of work.

Pros:
1. Theoretical analysis and empirical improvement of the Straight-Through estimator is an important avenue of work.
2. The paper makes a solid contribution of deriving the Fourier expansion of the Straight-Through estimator bias.
3. Based on this expansion, the paper proposes an algorithm with reduced bias. The algorithm is simple to implement, practical and appears to work slightly better than DARN.

Cons:
1. The key weakness of the theoretical part of the paper is that it focuses on the bias of the estimator, but does not quantify the variance, especially after the modifications. If reducing the bias was the only goal, one could use unbiased (but high-variance) estimators such as REINFORCE or VIMCO.
2. The final algorithm appears to be the DARN estimator combined with relaxation by uniform noise (“Bernoulli splitting uniform”) and scaling. The paper does not have an ablation showing how the uniform noise and scaling perform on their own.
3. There are a few incorrect statements that I’ve noticed.
* “As a side contribution, we show that the gradient estimator employed with DARN (Gregor et al., 2013), originally proposed for autoregressive models, is a strong baseline for gradient estimation.” - MuProp paper compared to this estimator under the name 1/2-estimator
* In Lemma 1 the “REINFORCE gradient” is just the exact gradient of the expectation, not a stochastic REINFORCE gradient.
* “ To the best of our knowledge, FouST is the first gradient estimate algorithm that can train very deep stochastic neural networks with Boolean latent variables.” This paper uses up to 11 latent variable layers, while [1] has trained models with >20 latent variable layers (although their “layers” have just one unit).
4. The derivation of “Bernoulli splitting uniform” trick is confusing and contains a lot of typos. For instance, the text before eqn. (14) implies that the distribution of u_i is U[-1, 1], which cannot be right and does not correspond to Algorithm 1. The statement that this trick does not lead to a relaxation is odd, since the function is being evaluated at non-discrete points.
5. There are generally many typos and some poor formatting in the math. For example, in eqn. (6) the coefficients are off by one: it should be c0 + c1 z1 + c2 z2^2 + … . The equations (10) and (11) are poorly formatted. The notation \partial_z1 f(u_1, u_2) in eqn. (14) is strange. In many places p^{i->½} is denoted as p^{1->½}.
5. I don’t think I understood the idea of representation scaling (Section 4.4). The eqn. (16) would suggest that the scaling should optimally be set to zero, which is just saying that the gradient is unbiased when the model does not use the latents. There is no other practical guidance on choosing this coefficient. Furthermore, one can always absorb the global scaling factor into the succeeding weights layer of the model, so this trick can probably be replaced by a modification of the weights initialization.
6. The experiments are missing a comparison to the Straight-Through Gumbel-Softmax estimator, introduced in the original Gumbel-Softmax paper. This is a popular biased estimator for Bernoulli latents, e.g. used in [1] [2]. Another interesting comparison would be [3] which proposes a lower-bias version of Gumbel-Softmax.
7. Figure 2 is missing the line for REBAR, even though this line is referred to on Page 8. Figure 2 and Figure 4 are both labeled as training ELBOs, despite the plots being different.

[1] Andreas Veit, Serge Belongie “Convolutional Networks with Adaptive Inference Graphs” ECCV 2018
[2] Patrick Chen, Si Si, Sanjiv Kumar, Yang Li, Cho-Jui Hsieh “Learning to Screen for Fast Softmax Inference on Large Vocabulary Neural Networks” ICLR 2019 https://openreview.net/forum?id=ByeMB3Act7
[3] Evgeny Andriyash, Arash Vahdat, Bill Macready “Improved Gradient-Based Optimization Over Discrete Distributions” https://arxiv.org/abs/1810.00116

**Experience Assessment:**

I have read many papers in this area.

**Review Assessment: Checking Correctness Of Derivations And Theory:**

I assessed the sensibility of the derivations and theory.

**Review Assessment: Checking Correctness Of Experiments:**

I assessed the sensibility of the experiments.

**Review Assessment: Thoroughness In Paper Reading:**

I read the paper at least twice and used my best judgement in assessing the paper.

---

> ### Author Response · Authors · 2019-11-11
> **Author Response**
>
> We thank you for finding our contributions solid and the theoretical analysis an important avenue of work. In the following we will try to address your criticism.
>
> > 1. The key weakness of the theoretical part of the paper is that it focuses on the bias of the estimator, but does not
> > quantify the variance, especially after the modifications. If reducing the bias was the only goal, one could use
> > unbiased (but high-variance) estimators such as REINFORCE or VIMCO.
>
> Response:
> As opposed to classical high-variance estimators such as REINFORCE, (high) variance is not essential to our estimator. It is controlled by hyperparameters and does not require specialized techniques such as incorporating control variates (as opposed to REINFORCE).
>
> Out of the 3 changes we suggest, uniform sampling is the only extra source of variance. We mention in the section on the ‘Bernoulli splitting’ trick, that in practice we do not sample from the full [0,1] or [-1,0] interval, but rather from some smaller [a,b] interval (chosen per task). The extra variance introduced by our method can be made small by making the size of the interval small. The interval sizes we chose in the experiments lie in the 0.01 to 0.2 range, as described in the paper.
>
>
> > 2. The final algorithm appears to be the DARN estimator combined with relaxation by uniform noise (“Bernoulli splitting uniform”)
> > and scaling. The paper does not have an ablation showing how the uniform noise and scaling perform on their own.
>
> Response:
> This is a valid concern and we have added ablation experiments to the paper (see appendix D.2, page 14). It should be kept in mind that comparing against DARN already justifies the combined effect of scaling and uniform noise to an extent.
>
> The ablation experiments were performed on the one stochastic layer non-linear OMNIGLOT and MNIST models to see the effect of scaling and uniform noise on importance weighted straight-through. The results show that scaling by itself helps in both cases. Uniform noise by itself improves performance on MNIST. Uniform noise on its own is worse on OMNIGLOT but improves performance when combined with scaling. The results were verified over multiple runs.
>
>
> > 3. There are a few incorrect statements that I’ve noticed.
>
> >* “As a side contribution, we show that the gradient estimator employed with DARN (Gregor et al., 2013), originally proposed for
> > autoregressive models, is a strong baseline for gradient estimation.” - MuProp paper compared to this estimator under the name
> > 1/2-estimator
>
> Response:
> It is true that some previous papers have used the DARN estimator in their experiments. On the other hand, the models used in such experiments have been small and the datasets tiny.
>
> Instead, our statement is intended in the context of larger models with many layers of latent variables. Ever since DARN's publication in 2013, DARN seems to have fallen into disuse after surveying the related literature. However, based on our theoretical analysis we were intrigued to find that DARN is in fact an effective and theoretically grounded estimator. Our experiments on larger and more complex models corroborate this finding. To this end, we feel that DARN deserves reconsideration by the community.
>
> > * In Lemma 1 the “REINFORCE gradient” is just the exact gradient of the expectation, not a stochastic REINFORCE gradient.
>
> Response:
> Yes, the lemma is about the exact gradient. The statement has been modified to make this clearer.
>
> > * “ To the best of our knowledge, FouST is the first gradient estimate algorithm that can train very deep stochastic neural networks
> > with Boolean latent variables.” This paper uses up to 11 latent variable layers, while [1] has trained models with >20 latent variable
> > layers (although their “layers” have just one unit).
>
> Response:
>      a) Stochastic variables in [1] are multiplicative and the gradient has a path through deterministic variables. See equation 3 in the referenced paper. This is different from the setting we consider, where we have functions of stochastic Boolean inputs only.
>
> In a similar vein, networks trained with Dropout also have multiplicative Boolean units (albeit with fixed parameters) leading to stochastic networks, but this doesn’t necessarily negate our claim.
>
>      b) It is much easier to train functions with very low stochastic dimension. In fact for a single variable our estimator is unbiased (or close to unbiased depending on chosen parameters).
>
> ... Continued in next comment.

---

> > ### Author Response · Authors · 2019-11-11
> > **Author Response [Continued]**
> >
> > > 4. The derivation of “Bernoulli splitting uniform” trick is confusing and contains a lot of typos. For instance, the text before
> > > eqn. (14) implies that the distribution of u_i is U[-1, 1], which cannot be right and does not correspond to Algorithm 1. The
> > > statement that this trick does not lead to a relaxation is odd, since the function is being evaluated at non-discrete points.
> >
> > Response:
> > u_i is not uniform over [-1,1] but is rather conditioned on the Bernoulli variable. It is uniform over U[0,1] if the Bernoulli variable is +1 and U[-1,0] if the Bernoulli variable is -1.
> >
> > The statement that this does not lead to a relaxation (of the sort in Gumbel-Softmax) refers to the fact that we still have a hard binary decision. By comparison, Gumbel-Softmax has no hard samples. Our modification can also be thought of as a function that given a Boolean input, takes  a uniform sample based only on the input as its first step. We have further clarified this in section 4.3 on page 6.
> >
> > > 5. There are generally many typos and some poor formatting in the math. For example, in eqn. (6) the coefficients are off
> > > by one: it should be c0 + c1 z1 + c2 z2^2 + … . The equations (10) and (11) are poorly formatted. The notation \partial_z1
> > > f(u_1, u_2) in eqn. (14) is strange. In many places p^{i->½} is denoted as p^{1->½}.
> >
> > Response:
> > Taking the instances one by one:
> >      a. There is no typo in equation 6. It gives the expression for the Fourier coefficient in terms of the Taylor coefficients rather than the Taylor expansion of f.
> >      b. We have fixed the formatting in equations 10 and 11.
> >      c. In the derivation notation \partial_z1 f(u_1, u_2) we think of z_1 as the input variable and u_i as samples to evaluate the function on.
> >      d. The p^{1->½} is used in section 4.3 in the illustration of a bivariate function. Here we take the gradient relative to the first variable and i is 1. A clarification has been added in section 4.3 on page 5.
> >
> > > 5. I don’t think I understood the idea of representation scaling (Section 4.4). The eqn. (16) would suggest that the scaling should
> > > optimally be set to zero, which is just saying that the gradient is unbiased when the model does not use the latents. There is no
> > > other practical guidance on choosing this coefficient. Furthermore, one can always absorb the global scaling factor into the
> > > succeeding weights layer of the model, so this trick can probably be replaced by a modification of the weights initialization.
> >
> > Response:
> > The intention in section 4.4 is to show that scaling affects the bias of the gradient by decaying higher-order Taylor coefficients. For simplicity we scaled activations. Other equivalent ways of scaling could be used.
> >
> > There is a trade-off involved here. If we scale too much we make the function close to deterministic which hurts distribution modeling. If we scale too little, this causes the bias to worsen.
> >
> > Deriving the optimal scaling, even more so adaptively, would be an interesting direction. For simplicity and to maintain focus and length, we resort here in simple hyperparameter tuning.
> >
> > > 6. The experiments are missing a comparison to the Straight-Through Gumbel-Softmax estimator, introduced in the original Gumbel-
> > > Softmax paper. This is a popular biased estimator for Bernoulli latents, e.g. used in [1] [2]. Another interesting comparison would be [3]
> > > which proposes a lower-bias version of Gumbel-Softmax.
> >
> > Response:
> > The Straight-Though Gumbel estimator is similar to the Gumbel estimator but underperforms it in terms of the optimization objective as reported in the original paper. It is used in cases where the task calls for hard binary samples.
> > Nevertheless, we have added comparisons to Straight-Through Gumbel in the updated paper (see figure 2 on page 8). Our results support the ones from the original paper that Gumbel-Straight Through performs worse in terms of the optimization objective than Gumbel Soft-max.
> >
> > ... Continued in next comment

---

> > > ### Author Response · Authors · 2019-11-11
> > > **Author Response [Continued]**
> > >
> > > > 7. Figure 2 is missing the line for REBAR, even though this line is referred to on Page 8. Figure 2 and Figure 4 are both labeled as
> > > > training ELBOs, despite the plots being different.
> > >
> > > Response:
> > > The reference on page 8 refers to the test values table in the appendix. We have made this clearer in section 7.2 on page 7.
> > >
> > > As far as comparing against REBAR is concerned, we believe that the estimators are not directly comparable for two reasons. First, REBAR can realistically be applied only to smaller networks with few stochastic layers. Second (also related to the first), it makes multiple function evaluations per input. It also makes multiple passes through the later layers of the network when there are multiple stochastic layers. The estimators that we compare against only make a single evaluation per input.
> > >
> > > Figure 2 and figure 4 report results on different datasets (OMNIGLOT and MNIST). Both report training ELBO’s.
> > >
> > >
> > > Thank you for the interest you have shown to our work and your points of improvement. We hope to have addressed persuasively your doubts that influence your score. We would be more than happy to explain further if more clarifications are needed.

---

> ### Comment · AnonReviewer2 · 2019-11-15
> **Reply to the response by the authors**
>
> I would like to thank the authors for their detailed reply to my review. The paper can certainly be improved, but, with the added ablations and clarifications, I feel that it would find its audience at ICLR. Hence, I am increasing my score to weak accept. However, I encourage the authors to work more on the text to increase the paper's impact.
>
> Comments on the response:
>
> > As opposed to classical high-variance estimators such as REINFORCE, (high) variance is not essential to our estimator.
>
> This claim is false in general. One can easily construct a function for which the REINFORCE gradient will have exactly zero variance, while the estimator you propose will have a non-zero variance. Consider univariate case and set f(z) = 1 / g_REINFORCE(z). Hence, you either have to prove that the method has lower variance, or demonstrate that the variance is empirically lower in the experiments.
>
> > uniform sampling is the only extra source of variance
>
> This is also not true in general, as importance sampling can also increase the variance of the estimator if the proposal distribution is not optimal.
>
> > This is a valid concern and we have added ablation experiments to the paper (see appendix D.2, page 14).
>
> Thank you, this ablation is very informative!
>
> > DARN estimator
> > “FouST is the first gradient estimate algorithm that can train very deep stochastic neural networks with Boolean latent variables”
>
> I agree with the revised/toned down version of your statements.
>
> > The statement that this does not lead to a relaxation (of the sort in Gumbel-Softmax) refers to the fact that we still have a hard binary decision.
>
> I still disagree: your method is clearly a relaxation since it evaluates the function at non-discrete points. What you are saying is that one can recover the hard sample from the relaxed sample. It is a good property, yet the networks can certainly exploit the extra noise injected by your method. This is especially misleading because you state in the introduction that “We resort to the term Boolean instead of binary to emphasize that we work directly on the Boolean space {−1, +1}, without any continuous relaxations or quantizations.”
>
> Related question to the authors: do you compute the ELBO in Table 1 and Figures 2-3 with Boolean samples? It is possible that the models are exploiting the extra Uniform noise.
>
> >      a. There is no typo in equation 6. It gives the expression for the Fourier coefficient in terms of the Taylor coefficients rather than the Taylor expansion of f.
> >      d. The p^{1->½} is used in section 4.3 in the illustration of a bivariate function. Here we take the gradient relative to the first variable and i is 1.
>
> Agreed, there are indeed no typos there. Thanks for the clarification.
>
> > we have added comparisons to Straight-Through Gumbel in the updated paper (see figure 2 on page 8). Our results support the ones from the original paper that Gumbel-Straight Through performs worse in terms of the optimization objective than Gumbel Soft-max.
> > REBAR
>
> Thank you for adding this comparison and the clarifications.

---

> > ### Author Response · Authors · 2019-11-15
> > **Further Clarification**
> >
> > Thanks very much for your response. The increased score is much appreciated.
> >
> > Further clarifications follow:
> >
> > > As opposed to classical high-variance estimators such as REINFORCE, (high) variance is not essential to our estimator.
> >
> > >>This claim is false in general. One can easily construct a function for which the REINFORCE gradient will have exactly zero
> > >>variance, while the estimator you propose will have a non-zero variance. Consider univariate case and set f(z) = 1 /
> > >>g_REINFORCE(z). Hence, you either have to prove that the method has lower variance, or demonstrate that the variance is
> > >>empirically lower in the experiments.
> >
> > While it is true that one can construct functions for which reparameterization or path-wise gradients have higher variance than REINFORCE, it is generally accepted that in the kind of models usually employed in the literature, REINFORCE gradients have high variance compared with reparameterization and Straight-Through gradients, necessitating the use of specialized variance reduction techniques. Our estimator is a version of the Straight-Through estimator and any extra variance we do add to Straight-Through is controlled with hyperparameters (unlike REINFORCE).
> >
> > Nevertheless, variance comparisons might still be useful and we will add such a comparison in future versions.
> >
> > > uniform sampling is the only extra source of variance
> >
> > >>This is also not true in general, as importance sampling can also increase the variance of the estimator if the proposal
> > >>distribution is not optimal.
> >
> > This statement should be related to the Straight-Through estimator in that we do not have any “extra” variance increasing step, apart from the one mentioned.
> >
> > As for the importance sampling, what we say is that the Straight-Through estimator can be interpreted as _missing_ an importance weighting coefficient. We do not perform any new importance sampling that could cause a further increase in variance.
> >
> > > The statement that this does not lead to a relaxation (of the sort in Gumbel-Softmax) refers to the fact that we still have a hard binary decision.
> >
> > >>I still disagree: your method is clearly a relaxation since it evaluates the function at non-discrete points. What you are saying is
> > >>that one can recover the hard sample from the relaxed sample. It is a good property, yet the networks can certainly exploit
> > >>the extra noise injected by your method. This is especially misleading because you state in the introduction that “We resort to
> > >>the term Boolean instead of binary to emphasize that we work directly on the Boolean space {−1, +1}, without any continuous
> > >>relaxations or quantizations.”
> >
> > >>Related question to the authors: do you compute the ELBO in Table 1 and Figures 2-3 with Boolean samples? It is possible
> > >>that the models are exploiting the extra Uniform noise.
> >
> > The intention here is to distinguish the estimator from continuous relaxation estimators. Continuous relaxation estimators replace a _discontinuous_ function by a _continuous_ function of its inputs. For example, Gumbel-Softmax replaces argmax by softmax.
> >
> > In our opinion, the kind of noise we add cannot be considered a continuous relaxation because we introduce no continuous functions.
> >
> > It is also unlikely that networks can use the extra uniform noise, by itself, to improve the training objective, since the noise is completely unstructured and unlearned. Unstructured noise usually results in slower training or possibly higher training error. An example of this would be dropout.
> >
> > Thanks again for the appreciation.

---

### Official Review · AnonReviewer3 · 2019-10-23
**Official Blind Review #3**

**Rating:** 3

**Review:**

------------- updated after rebuttal -------------------

I thank the authors for clarifying and correcting the notations in Lemma 3. Though I still think the current state of the derivation is presented in a suboptimal way, and as a result, can be misleading to people.

The Fourier analysis used to give the results that the exact gradient equals $2\hat{f}^{(p\to 1/2)}(i)$ (eq. 5) is totally unnecessary: Despite it might seem fancy as a Fourier coefficient, it is just another way of writing the local expectation estimator (Tokui, S., & Sato, I., 2017), if we expand it using the definition $\hat{f}^{(p)}(S) = E_{p(z)}[f(z)\phi_S(z)]$.

The authors argue that the Fourier analysis is essential to show the bias of the estimator. However, the only conclusion they draw from Fourier analysis is eq.5. And all the bias analysis follows by using Taylor expansions of it. The paper can be greatly simplified if they remove all boolean analysis parts and start from eq. 5 (which has a straightforward proof), using the conventional notation instead of Fourier coefficients.

During writing this, I read the bias correction section again and had another concern, the bias correction effect is only justified for functions with small mixed degree terms:

"For functions with small mixed degree terms, this can lead to bias
reduction, at the cost of an increased variance because of sampling an auxiliary variable"

For general multivariate functions, it is even not clear whether the proposed estimator has a smaller bias than the straight-through one. This weakness has a deep reason behind it because they are trying to generalize a bias reduction technique from a univariate function to multivariate functions, which, if done natively, would require K evaluations of the function (K is the number of input dimensions) (as I pointed out in the original review).

Overall, I argue rejecting the paper in its current form.

----------------------------------------------------------------

This is not my first time reviewing this paper. Previous concerns on clarity has been addressed and the paper is now more readable. Though I still believe that the boolean analysis part is unnecessary for deriving the final estimator (which can be easily derived from the exact local-expectation estimator E_p[f(z_i=1) - f(z_i=0)] and applying f(1) - f(0) = \int_0^1 f'(x) dx \approx f(e), e~Unif[0,1].) plus some importance sampling trick.

I think the proof of Lemma 3 is incorrect (though the conclusion is correct). f is never multi-linear in the continuous space. It is only for the boolean space, that f has an multi-linear form with Fourier expansion. So the claim that f is multi-linear then

E_{p(z}}[f(z_1, ..., z_n)] = f(\mu_1, ..., \mu_n)

is incorrect. This can only be true when f is also linear in the continuous space (which is not true for typical vaes).



**Experience Assessment:**

I have published one or two papers in this area.

**Review Assessment: Checking Correctness Of Derivations And Theory:**

I carefully checked the derivations and theory.

**Review Assessment: Checking Correctness Of Experiments:**

I assessed the sensibility of the experiments.

**Review Assessment: Thoroughness In Paper Reading:**

I read the paper thoroughly.

---

> ### Author Response · Authors · 2019-11-11
> **Author response**
>
> Thank you for appreciating the spirit of the work and the improved clarity.
>
> Reviewer 3 has raised two objections:
> 1. The estimator can be derived from a local-expectation estimator plus some importance sampling trick.
> 2. The proof of lemma 3 is incorrect since E[f(z)] \ne f(mu).
>
> Taking the second objection first. As stated in the proof of lemma 3 we are working with the Fourier representation of f which is a multilinear polynomial in the z_i and polynomials are defined for any input value. In VAE terms, the function f here is not the decoder of a VAE, but rather its Fourier representation. Given a decoder f:{-1,1}^n -> R, we use its Fourier representation (under the uniform Bernoulli distribution) which is a multilinear polynomial. For example for two variables the representation could be expressed as
> f(z_1,z_2) = c_0 + c_1 z_1 + c_2 z_2 + c_{1,2} z_1 z_2.
> Then E[f(z_1,z_2)] = c_0 + c_1 E[z_1] + c_2 E[z_2] + c_{1,2}E[z_1]E[z_2] = f(mu_1, mu_2), which is how it is intended in the proof of the lemma. Further clarification has been included in appendix B.1 on page 11.
>
> For the first objection, we give the following points:
>      1.  The point of the Boolean analysis is not just to derive the estimator but rather to express  the bias of the estimator analytically. This is the expression given in lemma 2. The importance sampling correction then follows from observation that it would reduce bias. The reviewer mentions applying the importance sampling trick, but this is not motivated by any analysis of why this would help with bias reduction.
> We believe a motivating analysis is important for the proposed importance sampling in order to identify the root cause. We believe the proposed harmonic analysis builds an effective framework for future bias reduction strategies.
>
>      2a. The estimator suggested by the reviewer in the univariate setting computes f’(u) where u is a uniform sample from [0,1]. This estimator does not use any Bernoulli samples. In our estimator we depend on Bernoulli samples followed by uniform samples conditioned by the Bernoulli samples. The two estimators are not the same even in the univariate case.
>
>      2b. The extension to the multivariate case is also different. The local-expectation method requires 2N evaluations of the function and results in an unbiased estimate. Our goal on the other hand is bias reduction rather than bias removal while maintaining efficiency. Consequently we make 1 function evaluation and justify that for appropriately behaving functions, the algorithm can reduce bias. For this purpose also, we need an analytical expression of the bias which is supplied by our analysis.
>
> Lastly, we would submit that a contribution of this work is the proposition that stochastic gradients and Fourier coefficients are identical. We feel that this contribution is of interest independent of its application in derivation of the estimator.
>
> We hope to have addressed persuasively your doubts that influence your score. We would be more than happy to further explain if you require further clarifications.

---

> > ### Comment · AnonReviewer3 · 2019-11-11
> > **on Lemma 3**
> >
> > Hi,
> >
> > I don't quite get your explanation.
> >
> > >  the function f here is not the decoder of a VAE, but rather its Fourier representation
> > isn't f and its Fourier expansion mathematically equal?
> >
> > > f(z_1,z_2) = c_0 + c_1 z_1 + c_2 z_2 + c_{1,2} z_1 z_2.
> > My point is that this equation only holds for $z \in \{-1,1\}^2$.
> > That's where f (a boolean function) is defined.
> > This equation doesn't hold in the continuous domain, right?

---

> > > ### Author Response · Authors · 2019-11-11
> > > **Clarification on the Lemma**
> > >
> > > Thanks very much for your question. We will try to elaborate further.
> > >
> > > > the function f here is not the decoder of a VAE, but rather its Fourier representation
> > > >> isn't f and its Fourier expansion mathematically equal?
> > >
> > > A Boolean function is defined only on the Boolean cube. The Fourier representation is defined everywhere. The two are guaranteed to be equal on the Boolean cube.
> > >
> > > > f(z_1,z_2) = c_0 + c_1 z_1 + c_2 z_2 + c_{1,2} z_1 z_2.
> > > >> My point is that this equation only holds for {1,-1}^2 . That's where f (a boolean function) is defined. This equation
> > > >> doesn't hold in the continuous domain, right?
> > >
> > > This equation _defines_ the Fourier representation of a Boolean function. Once we have the Fourier representation - a polynomial - we can evaluate it in the continuous domain.
> > >
> > > Furthermore, what the statement $E[f(z)] = f(\mu)$ says is that the expected value of the _Boolean_ function is equal to the Fourier representation evaluated at $\mu$. $E[f(z)]$, or $f(\mu)$, in general, is _not_ equal to the evaluation of a VAE decoder at $\mu$ and we never claim that they are equal. Evaluation of the Fourier representation on the continuous space leads to results about the Boolean function when only considering the Boolean cube as the input space.
> > >
> > > You can also see the proof of the lemma in the reference we provide. It is available online here.
> > > http://www.contrib.andrew.cmu.edu/~ryanod/?p=1339
> > > Please see equation 4 and the associated proof. The only difference is that the lemma in the reference has a single probability parameter, while we have a different parameter per input.
> > >
> > > We hope this is useful.

---

> > > > ### Comment · AnonReviewer3 · 2019-11-11
> > > > **confused by notations**
> > > >
> > > > Thanks. I see the point.
> > > > Maybe this is the convention in boolean analysis, but I do think that if the fourier expansion is supposed to be another function that is defined on the continuous domain, then a different notation from f should be used. That's how I'm confused by this equation.

---

> > > > > ### Author Response · Authors · 2019-11-12
> > > > > **Clarification Added**
> > > > >
> > > > > Yes, this is the conventional notation in Boolean analysis. To avoid confusion in the proof of the lemma, we now use a different notation for the representation in the updated version in appendix B.1 on page 11.
> > > > >
> > > > > Thank you for your comment. Please let us know if there are any other concerns or doubts.

---

### Official Review · AnonReviewer1 · 2019-10-24
**Official Blind Review #1**

**Rating:** 6

**Review:**

Summary:
The authors analyze the bias in the straight-through gradient estimator using the framework of harmonic analysis of boolean functions. Based on this analysis, they propose three methods to reduce the bias of the straight-through estimator, resulting in a less-biased estimator that is the same computational complexity as the original. They evaluate this estimator on a series of generative modeling tasks where they demonstrate improvements over existing methods, including the ability to train a very deep stochastic network.

I enjoyed this paper -- the exposition is clear, the ideas are (to my knowledge) novel and make sense, and the experimental evaluation is thorough and convincing. I recommend an accept.

I skimmed through the proofs in the appendix so cannot with absolute confidence vouch for their correctness.

One small piece of feedback: I found the most confusing part of the paper was the section on the 'bernoulli splitting trick'. It might be helpful to pull some of the appendix material into this section to make it a little less sparse.


**Experience Assessment:**

I have published one or two papers in this area.

**Review Assessment: Checking Correctness Of Derivations And Theory:**

I assessed the sensibility of the derivations and theory.

**Review Assessment: Checking Correctness Of Experiments:**

I carefully checked the experiments.

**Review Assessment: Thoroughness In Paper Reading:**

I read the paper thoroughly.

---

> ### Author Response · Authors · 2019-11-11
> **Author response**
>
> We thank the reviewer for the encouraging review and for appreciating the novelty and thoroughness of the experiments. With regard to the suggestion of clarifications in section 4.3, we have incorporated this in the updated version (section 4.3, page 5). We hope to have made the matter clearer.

---

### Decision · Program_Chairs · 2019-12-19

**Decision:**

Reject

**Comment:**

Straight-Through is a popular, yet not theoretically well-understood, biased gradient estimator for Bernoulli random variables. The low variance of this estimator makes it a highly useful tool for training large-scale models with binary latents. However, the bias of this estimator may cause divergence in training, which is a significant practical issue. The paper develops a Fourier analysis of the Straight-Through estimator and provides an expression for the bias of the estimator in terms of the Fourier coefficients of the considered function.

The paper in its current form is not good enough for publication, and the reviewers believe that the paper contains significant mistakes when deriving the estimator. Furthermore, the Fourier analysis seems unnecessary.